# Inclusion Complex of Docetaxel with Sulfobutyl Ether β-Cyclodextrin: Preparation, In Vitro Cytotoxicity and In Vivo Safety

**DOI:** 10.3390/polym12102336

**Published:** 2020-10-13

**Authors:** Lili Ren, Xiaolong Yang, Weilu Guo, Jin Wang, Guoguang Chen

**Affiliations:** 1School of Pharmacy, Nanjing Tech University, 5th Mofan Road, Nanjing 210094, China; renlili@njtech.edu.cn (L.R.); cresc654123@163.com (X.Y.); guowl726@163.com (W.G.); wjnjtech@163.com (J.W.); 2Department of Microbiology and Immunology, Stanford University, Stanford, CA 94305, USA

**Keywords:** inclusion complex, docetaxel, bioavailability, cytotoxicity

## Abstract

Docetaxel (DTX), as a first-line anti-tumor drug, has been studied for decades for its diverse bioactivities. However, DTX presents poor solubility in water, low bioavailability and serious toxic side effects which has hindered its application in the clinic. To address these problems, docetaxel-sulfobutyl ether-β-cyclodextrin inclusion complex (DTX-SBE-β-CD) was prepared successfully by saturated aqueous solution method. Sulfobutyl ether β-cyclodetrin (SBE-β-CD) is used as delivery material. For this study, the inclusion complex of docetaxel with sulfobutyl ether β-cyclodetrin (DTX-SBE-β-CD) was prepared and optimized its properties to enhance the cytotoxicity of cancer cells. A large number of physical characterization results showed that DTX-SBE-β-CD inclusion complex was successfully prepared by saturated aqueous solution method. DTX-SBE-β-CD inclusion complex was optimized by Central Composite Design. DTX-SBE-β-CD had an inhibitory effect on the in vitro determination of MCF-7 and HepG2 cells by MTT assay. Pharmacokinetic studies were carried out on male Sprague–Dawley rats by tail injection, including the distribution, metabolism and elimination of DTX-SBE-β-CD in vivo. In the experimental study of inhibition of cancer cells, DTX and DTX-SBE-β-CD showed apparent concentration-dependent inhibitory actions on tumor cells and the inhibition of DTX-SBE-β-CD group was more obvious.

## 1. Introduction

With the invention of high-throughput screening methods, the percentage of new active molecules that are insoluble in water has risen to about 40% in total; in various therapeutic areas this percentage has even reached 80–90%. Those active molecules are hydrophobic compounds, which exhibit low absorption and transportion properties. Though they have high pharmacological activity, they are facing exclusion due to their low water solubility and bioavailability. Hence, pharmaceutical scientists have responded to this challenge by developing a variety of formulation principles for these poorly water-soluble drugs for clinical applications. Insoluble drugs should be made dissolvable by physico-chemical or biological means in order to arrive at the pharmacological target in appreciable amounts. Various approaches exemplified by the addition of co-solvents [1,2,3], solid dispersion [4], size reduction [5,6], as well as the complexation with cyclodextrins(CD) [7], have been used to optimize the solubility of insoluble substances.

Cyclodextrin is a cyclic oligosaccharide formed by 6 (α-CD), 7 (β-CD) or 8 (γ-D) (α-1,4)-linked D-glucopyranose units [8]. Cyclodextrin has outer hydrophilic and inner hydrophobic characteristics, so it can form complexes with molecules of a certain size and shape [9,10,11]. Those molecules can enter entirely or partly replacing the high-energy water molecules from the interior cavity. So it may affect the encapsulated molecule physicochemical characteristics, improving drug solubility and bioavailability. As the use of cyclodextrins is restricted by low its aqueous solubility [12], derivatives of cyclodextrin with better properties have been developed, including 2-hydroxypropyl-β-cyclodextrin (HP-β-CD), methyl-β-cyclodextrin (M-β-CD), carboxymethyl-β-cyclodextrin (M-β-CD), sulfobutyl-ether-β-cyclodextrin (SBE-β-CD), etc. Among them, SBE7-β-CD (its trade name is Captisol^®^), a chemically modified β-cyclodextrin was patent protected by CyDex Corporation. SBE-β-CD is non-toxic and biocompatible CD derivatives which exhibits solubility and complexing abilities greater than those of the parent β-CD. Meanwhile, the four-carbon butyl chain of SBE-β-CD coupled with the repulsion of the end group’s negative charge gives an extremely hydrophilic exterior surface and an extended hydrophobic cavity, which compared with other CD derivatives [13,14].

Docetaxel (DTX), belonging to taxanes, preferentially binds to microtubuloproteins, stabilizing the microtubules and inhibiting division, thus causing cell cycle termination and eventual cell death. Docetaxel has presented a broad spectrum of activity against various kinds of tumors. Because of its poor aqueous solubility (1–5 μg·mL^−1^ in plain water), the commercial docetaxel (Taxoter^®^) is currently formulated in a combination of tween 80 and ethanol to improve it solubility [15,16,17]. Studies have shown that preparations are known to cause severe peripheral neuropathy and allergic reactions. Moreover, after dilution with ethanol solution, its physical properties are unstable and it needs to be used within 8 h; additionally, the side effects might be intensified. In order to increase the solubility and effectively protect docetaxel from degradation, multiple coupling compounds or nanoparticles are often used as solubilizers and transporters. The proven docetaxel formulations include dendritic macromolecules [18,19], low molecular weight chitosan [20,21,22] C60 fullerenes [23], fat-based formulations [24] and gold nanoparticles [25,26]. Although there are optimizations for drug distribution and tumor killing efficiency in organisms, the inherent polydispersity of these systems affects their structure–activity relationships [27,28,29]. The ideal docetaxel preparation requires a clear molecular structure and excellent biocompatibility, high drug loading and moderate drug release feature to avoid ineffective and side effects.

In this study, we prepared the inclusion complex and the experimental conditions were optimized to obtain the highest inclusion efficiency. Specifically, we chose docetaxel (DTX), an insoluble chemotherapeutic drug with severe toxicities [30], as a model payload to synthesize docetaxel-sulfobutyl ether-β-cyclodextrin inclusion complex (DTX-SBE-β-CD). The cytotoxicity of DTX-SBE-β-CD in vitro was evaluated on cancer cells using MTT assay. The pharmacokinetic study was also conducted to analyze the pharmacokinetic characteristics of DTX-SBE-β-CD in vivo. We expected that this strategy induces an added inhibitory effect on tumor growth with increased efficiency and low toxicity.

## 2. Experimental

### 2.1. Materials

DTX was obtained from Jiangsu Nhwa pharmaceutical Co., Ltd. (Xuzhou, China). SBE-β-CD (Mw = 2.24 kD, mean substitution degree of 7) was purchased from Nanjing JuHuan Medical Technology Co., (Nanjing, China) Ltd. and bovine serum was purchased from Shanghai Luoshen Biotechnology Co., Ltd. (Shanghai, China). Tryptose was obtained from Beyotime Biotechnology Co., Ltd. Anhydrous ethanol was obtained from Sinopharm Group Chemical Reagent Co., Ltd. (Shanghai, China). Acetonitrile was obtained from Sinopharm Group Chemical Reagent Co., Ltd. (Shanghai, China). Methanol was obtained from Sinopharm Group Chemical Reagent Co., Ltd. (Shanghai, China). Tert-butyl ether was obtained from Sinopharm Group Chemical Reagent Co., Ltd. (Shanghai, China).

Instrument: BP211D Cedris electronic balance Sedris GMBH, Gottingen, Germany; Kq-300db CNC ultrasonic cleaner Kunshan Ultrasonic Instrument Co. LTD, Kunshan, China; Model 85-2 constant temperature magnetic stirrer Jiangsu Jintan Medical Instrument Factory, Nanjing, China; Dsc-204 thermal analyzer Germany Nach Company, Bavaria, Germany; Infrared spectrometer Fourier (Tianjin) FTIR Corporation, Tianjin, China; High performance liquid chromatography Diane (China) Co., LTD, Shanghai, China; Lgj-10 freeze dryer Beijing Sihuan Scientific Instrument Factory, Beijing, China.

### 2.2. Methods

#### 2.2.1. Cell lines and Cell Culture

The cells strains of human breast cancer cell MCF-7 and human hepatoma cell HepG2 were all obtained from Shanghai Cell Institution. The strains of the two cancer cells were conventionally cultured in RPMI-1640 medium supplemented with 10% fetal bovine serum in a weak alkaline condition (pH: 7.2–7.4). These cells were put under the condition of 37 °C, 5% carbon dioxide purge until used.

#### 2.2.2. Chromatographic Conditions

A Shimazu HPLC equipped with a LC-20A pump and a PDA detector was used for the analysis of the concentration of DTX. The detection process was performed on a Kromasil C18 column (5 μm, 250 mm × 4.6 mm). The mobile phase is consisting of distilled water/methanol (76:24), and its flow rate was determined at 1.0 mL/min for isocratic elution. Column temperature was set at 25 °C. Detective wavelength was set at 230 nm based on the previous research by Kim et al. [31]. The inclusion rate was calculated according to the HPLC method. The optimum molar ratio of Docetaxel and SBE-β-CD were determined according to the maximum inclusion rate.

#### 2.2.3. Preparation of DTX-SBE-β-CD Inclusion Complex

Saturated aqueous solution method were used in the preparation of DTX-SBE-β-CD inclusion complex. Docetaxel (40 mg) was dissolved in ethanol (20 mL) to obtain the solution of 2.0 × 10^−3^ mol·L^−1^. Weighing SBE-β-CD according to different mole ratio of Docetaxel and SBE-β-CD (molar ratio of DTX to SBE-β-CD ranging from 1:14 to 1:1), dissolving it in the solution of Docetaxel (1 mL, 2.0 × 10^−3^ mol·L^−1^), stirring at room temperature for 1.5 h and decompression spin dry. The product was dissolved by water and filtered. The filtrate was analyzed by HPLC to calculate the encapsulation rate.

#### 2.2.4. Optimized Formulation of DTX-SBE-β-CD by Central Composite Design

Based on the above study, a molar ratio of DTX and SBE-β-CD was determined as the optimal composition in the preparation of the inclusion complex. The factors affecting the preparation of inclusion compound are the dosage of cyclodextrin, stirring rate, temperature and reaction time. reaction time (A), temperature (B) and stiration rate (C) were chosen as experimental factors which could be optimized by Central Composite Design. Drug-loading rate was chosen as the criterion.

According to the technological conditions designed by orthogonal experiment, DTX-SBE-βCD inclusion compound was prepared by solution stirring under 9 technological conditions, and the solid powder was dried and cooled for further use. A certain amount of inclusion compound was precisely weighed and ultrasonic dissolved in distilled water was used for filtration. A proper amount of the added filtrate (1 mL) was taken into a 10-mL measuring bottle, and the mobile phase was added to the scale. The sample was shaken and injected with 20 L, and chromatogram was recorded. According to the method under “Section 2.2.2”, the peak area A was substituted into the above standard curve equation to obtain the solubility of docetaxel.

#### 2.2.5. Preparation of DTX-SBE-β-CD Inclusion Freeze-Dried Preparation

Lyophilization is the best way to stabilize the inclusion compound due to the poor physical and chemical stability of the inclusion solution. Therefore, DTX-SBE-β-CD clathrate is freeze-dried, in which, SBE-β-CD is used as the drug carrier and the support materials for freeze-dried products.

##### Determination of Freeze-Drying Conditions

According to the basic principle of freeze-drying and the material condition in the freeze-drying process, the best freeze-drying technology was determined and three batches of finished products were prepared according to this method.

##### Appearance Evaluation

The appearance should not collapse or shrink, should remain full and maintain the original volume; the whole piece can fall off without the phenomenon of fragmentation. Color and luster should be uniform, with no spots, fine and loose porous.

##### Redispersibility Evaluation

Take a bottle of DTX-SBE-β-CD inclusion lyophilized preparation, add 2 mL distilled water and shake. After shaking, observe whether the solid can disperse quickly to get a clear and transparent solution.

#### 2.2.6. Phase-Solubility Studies

In this study, the stability constant was assessed by classic determination method using solubility diagrams that required calculations involving drug solubility [32]. Briefly, an excess amount of DTX was added to the aqueous solution containing 0–20% (*w*/*v*) SBE-β-CD, sonicated at room temperature for 5 min in sealed glass vials. Then the resulting suspensions were shaken in a thermostatic air-bath shaker at 25 °C for 48 h until equilibrium was achieved. After that, the saturated supernatant fluid was centrifuged for 5 min under 5000 rpm, and then the top fluid was withdrawn and filtered through a 0.45-μm filter. The filtrate was diluted appropriately with deionized water and analyzed by HPLC as reported above.

### 2.3. Characterization

The inclusion complex of DTX with SBE-β-CD was investigated through the following methods. DTX, SBE-β-CD and their physical mixture (DTX: SBE-β-CD = 1:12, molar ratio) were also characterized for comparison in order to determine whether the inclusion complex was formed.

#### 2.3.1. Infrared Fourier Transformation (FT-IR)

FT-IR was recorded by a Fourier transform infrared spectrometer (MAGAN-IR 560 spectrometer) in the framework region of 4000–400 cm^−1^. Docetaxel (10 mg), SBE-β-CD, DTX-SBE-β-CD and their physical mixture were prepared as KBr disks at room temperature.

#### 2.3.2. H NMR

^1^H NMR spectra of docetaxel, SBE-β-CD, DTX-SBE-β-CD inclusion complex and their physical mixture were obtained on AV-500 400MHz (Bruker, Switzerland). All samples used were dissolved in 99.98% DMSO and filtered before use.

#### 2.3.3. Different Scanning Calorimetry (DSC)

The thermal behavior of docetaxel, SBE-β-CD, DTX-SBE-β-CD inclusion complex and their physical mixture was studied by DSC (NETZSCH, Germany). Samples were sealed in aluminum pans and heated from 30 to 380 °C at heating rate of 10 °C/min in an atmosphere of nitrogen.

#### 2.3.4. X-ray Diffraction Analysis (XRD)

The X-ray diffraction patterns of SBE-β-CD, docetaxel, their physical mixture and DTX-SBE-β-CD inclusion complex were carried out using a Rigaku Powder X-ray diffraction system (Model D/MAX-2200 Ultima/PC, Japan) with Ni-filtered Cu-Kαradiation powered at 40 kV and 30 mA. The samples were packed into a quartz cell of 0.2 cm depth and analyzed over an angular range from 2–40° in continuous scan mode by using a step size of 0.02° and a scan rate of 10°/min.

#### 2.3.5. Scanning Electron Microscopy (SEM)

Docetaxel, SBE-β-CD, DTX-SBE-β-CD inclusion complex and their physical mixture were examined by a scanning electron microscope (ISM-6510, Japan). Each sample was placed on an aluminum stub by conductive adhesive and sputter coated with a thin layer of gold in a vacuum for 120 s (Hitachi E-1010, Japan). The pictures were taken at an electric voltage 15 kV under different magnifications.

### 2.4. Cytotoxicity Assay

To evaluate whether inclusion compounds are toxic to cells, we chose MCF-7 and HepG2 as subjects and used different concentrations of drugs for toxicity experiments. Different concentrations of DTX, SBE-β-CD and DTX-SBE-β-CD (1, 5, 10, 20, 50 μg·mL^−1^) were prepared using gradient dilution method with culture medium (RPMI-1640 medium with 10% fetal bovine serum).

The suspension of MCF-7 and HepG2 cells at logarithmic growth were seeded into the 96-well plate (cell density was adjusted to 1 × 10^4^ cells per well) and exposed to different concentrations of DTX, SBE-β-CD and DTX-SBE-β-CD. The blank group was pure culture medium. The control group was given no drug but culture medium. The cells would be cultured for 24 h at the same condition as before (37 °C, 5% CO_2_). Twenty-four hours later, the RPMI-1640 medium was replaced by 10 µL MTT (3-[4-dimethylthiazol-2-yl]-2, 5- diphenyltetrazolium bromide) and 90 µL fresh culture medium and the cells would be cultured for another 4 h. After the incubation, 100 µL DMSO was added into each well and allowed to complex in a shaker for 10 min until complete dissolution of the formazan crystallization. StatFAx-2100 ELISA Reader was then used to detect the absorbance of the DMSO-cells mixed solution at the wavelength of 490 nm, and the detection was parallelly conducted in triplicate, the average value was used in the calculation of cell survival rate.

### 2.5. Pharmacokinetic Study

The in vivo pharmacokinetic study in this paper was approved by the animal protection and use committee of Nanjing medical university and followed the ‘‘Principles of Laboratory Animal Care’’.

Docetaxel API (Active Pharmaceutical Ingredients) and DTX-SBE-β-CD (10 mg/kg) inclusion complexes were both tested with 12 male Sprague–Dawley rats, randomly divided into two groups and the rats were allowed to have access to water and standard food. Blood samples (0.5 mL) were taken from the retinal vein of the rats using a capillary and at the following time points after injecting: 5, 10, 15, 30 min, 1, 2, 4, 6, 8, 10 and 12 h. The sample should be transferred to the heparin sodium tube for anticoagulation. The samples were centrifuged at 4000 rpm for 10 min. The resultant plasma was poured into 2-mL cryogenic plastic tubes and stored in a freezer (at 20 °C) until assayed. A week later, the two groups of animals exchanged the drug and obtained plasma samples, which were stored in the −20-degree refrigerator for analysis. The sample was extracted in the following steps: the plasma (50 mL) was mixed with 10 mL of paclitaxel solution (4.0 mg/mL) as a base, and 1.0 mL methyl tert-butyl ether in a 2-mL plastic centrifuge tube, centrifuged at 10,000 rpm for 5 min to precipitate the proteins and the supernatant layer (0.8 mL) was blown dry by N_2_ and is dissolved by 0.1 mL methanol, which was centrifuged at 10,000 rpm for 10 min. The supernatant layer (80 mL) was analyzed by LC-MS (Waters, American).

### 2.6. Data Analysis and Statistics

The plasma concentration versus time curves of the two preparations were plotted. The pharmacokinetic parameters such as elimination rate (Ke) and elimination half-life (T1/2) were calculated by using DAS2.0 software. The maximum peak concentration (Cmax) and the corresponding sampling time (Tmax) were obtained from the curves directly. The areas under the plasma concentration–time curve from time zero to time t (AUC0–t) and from time zero to infinity (AUC0-∞) were calculated by the method of moments. Statistical comparisons were made by using a ANOVA test and p value smaller than 0.05 was considered statistically significant.

### 2.7. Biodistribution Study

To investigate the distribution of DTX and DTX-SBE-β-CD in the normal rats 10, 60 and 360 min after administration (10 mg/kg), male rats were euthanized and their hearts, livers, spleens, lungs, kidneys, brains, stomaches, intestines, muscles, ovaries or testes, fat and marrows were collected. The organs were homogenized in 1 mL of PBS, and analyzed by LC-MS/MS to quantify the concentration of DTX.

### 2.8. Safety Study

#### 2.8.1. Hemolytic Test

Fresh blood 15–25 mL from a New Zealand White Rabbit was stirred to remove fibrin. Then, right amount of saline was added and subsequently centrifuged at 2000 r/min for 10 min to remove supernatant. Whole blood was collected and diluted to 100 mL with normal saline to prepare 2% erythrocyte suspension.

The test tubes numbered 1–6 were successively added with 2.5 mL 2% red blood cell suspension and different amounts of saline according to the Table 1, and tube 7 was added with 2.5 mL 2% rabbit red blood cell suspension and 2.5 mL distilled water. Different amounts of docetaxel formulation solution was added after water bath at 37 °C for 30 min. It was observed whether the solution had hemolysis at 15, 30, 15 min and 1, 2, 3, 4 h.

#### 2.8.2. Vascular Stimulation Test

Healthy New Zealand white rabbits were randomly divided into 3 groups (*n* = 3) and received a single ear vein injection of DTX, saline and DTX-SBE-β-CD with a dosage of 0.7 mg/kg. Preparations were administered every day for a total of three injections. At designed time points, rabbits were anesthetized and ear tissue was soaked with 10% formalin. Then the histopathologic examination was conducted.

## 3. Results and Discussions

### 3.1. Study of Mole Ratio

As can be seen from Figure 1, the inclusion ratio started to tend to be constant and the dissolution was good, when the ratio of the input ratio of docetaxel to SBE-β-CD was 1:12.

### 3.2. Optimized Formulation of DTX-SBE-β-CD by Central Composite Design

As shown in Table 2, a CCD (Central Composite Design) with three levels (−1,0,1) and three factors formed 20 experiments, the range of A were 1–3 h, B was 25–45 °C and C was 500–900 rpm. The experiment range of each variable was set based on the result of precious experiments. Design-Expert.V8.0.6.1 (Stat-Ease, Minneapolis American) software was utilized to design and analyze the experimental data. The correlation coefficient was 0.94. The polynomial regression equations of A, B and C on were represented as follows:Drug-loading rate = −24.97 + 2.81A + 0.97B + 0.029C + 5.5 × 10^−3^AB−1.25 × 10^−4^AC + 1.37 × 10^−5^BC − 0.73A^2^ − 0.014B^2^ − 2.12 × 10^−5^C^2^(1)

The contour and 3D surface plots of factors and yield were drawn in Figure 2a–c. The reaction time and stiration rate were kept stable, and temperature was mutative, when the temperature increased, the drug-loading rate increased first and subsequently decreased. The temperature was kept stable, reaction time and stiration rate were mutative, when reaction time or stiration rate increased, the drug-loading rate increased first all and then subsequently decreased. To maximize drug-loading rate, Design-Expert.V8.0.6.1 (Stat-Ease, Minneapolis American) was used to select the optimal conditions.

The optimum experimental parameters can be obtained according to the calculation results of Central Composite Design. The inclusion temperature of 35.4 °C, inclusion time of 2.97 h and stirring rate of 695 rpm were selected.

### 3.3. Preparation of DTX-SBE-β-CD inclusion Freeze-Dried Preparation

#### 3.3.1. Preparation Methods

Preliminary experimental results showed that the freeze drying process was: under the optimal process conditions of DTX-SBE-β-CD clathrate aqueous solution, 3-mL bottle packing into ampoules, keeping freeze-dried samples in freeze-drying box, inserting the temperature sensor, opening the lyophilizer. When the temperature dropped to −40 °C, the vacuum pump was turned on and the temperature was gradually increased by 2–4 °C per hour to sublimate the drying. When the temperature reached 25 °C, samples were insulated for 5 h before downtime when they were plugged and deflated out of the box and covered. Continued preparation of 3 batches was performed.

#### 3.3.2. Appearance Character

The surface of the lyophilized preparation is smooth, with no specks, uniform color and can be peeled off as a whole.

#### 3.3.3. Redispersibility Evaluation

A clear and transparent solution was obtained by taking 1 bottle of DTX-SBE-β-CD clathrate lyophilized preparation and adding 5 mL distilled water to vibrate.

### 3.4. Phase-Solubility Studies

The phase-solubility diagram of DTX in SBE-β-CD aqueous solution was shown in Figure 3. It could be seen that, the aqueous solubility of DTX linearly increased with the SBE-β-CD concentration (r = 0.9935), which could be defined as classic A_L_-type of Higuchi. As expected, the DTX and SBE-β-CD formed as well as dissolved inclusion complex with the molar ratio of 1:1. The apparent stability constant (K_1:1_) was calculated from the slope of the linear phase-solubility profile according to the equation. K_1:1_ = slope/S_0_ (1−slope), where S_0_ is the intrinsic solubility of the drug DTX [33]. The K_1:1_ value for DTX/SBE-β-CD was found to be 127.6 M^−1^ under 25 °C, and the water solubility of DTX was about 800 times higher than its original values. Thus, SBE-β-CD could significantly improve the water solubility of DTX.

### 3.5. Characterization

#### 3.5.1. Infrared Fourier Transformation (FT-IR)

Pressed together with KBr, the infrared spectra of docetaxel, SBE-β-CD, physical mixtures and DTX -SBE-β-CD inclusions were measured; results are shown in Figure 4. As can be seen from the figures above, the stretching vibration of 1737 and 1712 cm^−1^ in SBE-β-CD is C=O, indicating that the structure of docetaxel contains-C=O; the stretch of –C=O at 1712 cm^−1^ shifted to the low-frequency, which indicates that the molecular structure contains unsaturated O=C–C=C. 1269 and 1070 cm^−1 1^: the flexural vibration of the -C-O in the ester. The 978, 799, 778 and 709 cm^−1^ in SBE-β-CD were the vibration of =CH. 978 cm^−1^ was caused by the bending vibration of =CH and 799, 709, 778 cm^−1^ were caused by the =CH of conjugated dienes (or polyene). As docetaxel contains a benzene ring structure, 1497 cm^−1^ belonged to the benzene ring. The map of the physical mixtures was the superposition of the docetaxel and SBE-β-CD spectra. As the figures show, some characteristic peaks of SBE-β-CD still exist, however the characteristic peaks of docetaxel at 1730–1600, 1350–1000 and 750–650 cm^−1^ reduced or even disappeared, which proved that docetaxel entered the cavity of SBE-β-CD and formed the clathrate.

#### 3.5.2. H NMR

In Figure 5, the peaks at δ1.103 and δ1.235 (A) belonged to the two methyl of C16 and C17 in docetaxel and SBE-β-CD had no signals here, the physical mixture of docetaxel and SBE-β-CD appeared the same signal at δ1.103 and δ1.235, while the peaks of DTX-SBE-β-CD split at the same place; this fully showed that the two methyl groups of C16 and 17 interacted with the hydrogen of SBE-β-CD, docetaxel entered the cavity of SBE-β-CD and Docetaxel-SBE-β-CD inclusion compound formed.

#### 3.5.3. Different Scanning Calorimetry (DSC)

As can be seen from the Figure 6, the heat absorption peak of 215 °C was the melting point of the docetaxel, indicating that the melting point of docetaxel was at this temperature; in curve B, SBE-β-CD was the characteristic peak of the cyclodextrin in 274.5 of the absorption peak, namely the melting decomposition peak. Curve C was the physical mixture of docetaxel and SBE-β-CD, and there was no new phase, and the curve can be seen as the superposition of two material characteristic curves. For curve D, the melting point of docetaxel did not appear and the SBE-β-CD molten decomposition peak was almost not visible. Docetaxel and the physical mixture of docetaxel and SBE-β-CD is present, which can diagnose that the crystal type of docetaxel had changed and docetaxel came into the cavity of SBE-β-CD and formed the new phase, so the characteristic peaks of docetaxel and SBE-β-CD disappeared.

#### 3.5.4. X-ray Diffraction Analysis (XRD)

The XRD spectra of SBE-β-CD, docetaxel, their physical mixture and DTX-SBE-β-CD inclusion complex are shown in Figure 7. The diffractogram of docetaxel displayed numerous characteristic peaks due to its self-lattice arrangement, which indicated the crystallinity of the drug. In contrast, absence of any characteristic peak in the spectrum of SBE-β-CD indicated its amorphous state. The diffraction pattern of the physical mixture was just the superposition of SBE-β-CD and docetaxel. This may indicate that there was no interaction between SBE-β-CD and docetaxel. Compared with the diffractogram of pure SBE-β-CD and docetaxel, the diffraction pattern of the inclusion complex was overlapped with that of the SBE-β-CD, and obviously showed no characteristic peaks that pure docetaxel had. These results indicated that the self-lattice arrangement of docetaxel was changed from crystalline to amorphous state, which might be attributed to the docetaxel inclusion into the SBE-β-CD cavity.

#### 3.5.5. Scanning Electron Microscopy (SEM)

The combination of the physical mixture of docetaxel, SBE-β-CD, docetaxel and SBE-β-CD, and the inclusion of docetaxel-SBE-β-CD were observed under the electron scanning microscope. As shown in Figure 8.

B was SBE-β-CD, which had a border round spherical and was smooth; A was docetaxel with clear columnar crystal edges and corners; C was the physical mixture with both mechanical mixing, and the two forms were basically unchanged. As can be seen from D, the whole appeared as functional branches; with SBE-β-CD and docetaxel, the size increased slightly and it was also found that smaller particles’ adhesion to the larger particles, had been hard to find in this match and SBE-β-CD monomer.

### 3.6. In Vitro Cytotoxicity

The survival rates of DTX SBE-β-CD and DTX- SBE-β-CD on MCF-7 and HepG2 cells at different concentrations were displayed in Figure 9. Both DTX and its inclusion complex showed significant inhibition on MCF-7 and HepG2 cells while it should be noticed that DTX- SBE-β-CD showed greater inhibition compared to DTX and SBE-β-CD. For DTX- SBE-β-CD, when the concentration increased to 50 μg·mL^−1^, the survival rate on MCF-7 and HepG2 cells were all below 15% and lowest survival rate (<10%) were achieved on HepG2.

### 3.7. Pharmacokinetic Study

The mean docetaxel concentration versus time profiles following injection of docetaxel API and DTX-SBE-β-CD to Spragu–-Dawley rats were shown in Figure 10.

The corresponding mean pharmacokinetic parameters for docetaxel API and DTX- SBE-β-CD are tabulated in Table 3. As the date shown, the plasma concentration of docetaxel API and DTX-SBE-β-CD all reached the maximum immediately and then decreased rapidly. The Cmax of DTX- SBE-β-CD was 2049.2 ± 147.4 ng/mL, which was 1.8 times the docetaxel API. The AUC (0-t) and AUC (0-∞) for DTX- SBE-β-CD were also higher than docetaxel API, which were almost double compared to docetaxel API. The relative bioavailability of DTX-SBE-β-CD was 183.8%. The results revealed that the formulation of docetaxel as a clathrate made with SBE-β-CD significantly promoted and sustained the absorption of docetaxel.

### 3.8. Biodistribution Study

To investigate the in vivo distribution of DTX and DTX-SBE-β-CD, the DTX content of major organs and tissues were measured 10, 60 and 360 min after administration (Figure 11). At first, the drug quickly distributed in liver, spleen and kidney after administration, however, the distribution in marrow, lung and spleen at 60 and 360 min post injection for both drugs was significantly elevated. The concentration of DTX in brain, fat and testis was low throughout the experiment. As shown in Figure 12, the distribution of two formulations in organs or tissues was almost the same and the changing trend was similar.

### 3.9. Safety Study

#### 3.9.1. Hemolysis Test

DTX-SBE-β-CD solution had no hemolytic reaction to red blood cells, but distilled water had a hemolytic reaction (Table 4).

#### 3.9.2. Vascular Stimulation Test

Injection pinhole of saline and DTX-SBE-β-CD groups healed well without exudation, swelling, etc., and the injection site hardly had an obvious pathological reaction. However, DTX group developed serious vascular inflammation and surrounding tissue lesions (Figure 12).

## 4. Conclusions

In this study, DTX-SBE-β-CD inclusion complex was successfully prepared by using the saturated aqueous solution method. The FT-IR, DSC, XRD, DSC and SEM characteristics showed that the inclusion complex was formed. In addition, according to the data of H NMR, the two methyl groups of C16 and C17 in docetaxel molecules inserted in the cavity of SBE-β-CD could be the most likely complexation pathways. Besides, the in-vitro dissolution of docetaxel from the inclusion complex was significantly higher than that from docetaxel.

Additionally, the result of the MTT assay indicated that the DTX-SBE-β-CD had a more obvious inhibitory effect on the in vitro determination of MCF-7 and HepG2 cells compared with docetaxel and its physical mixture. Pharmacokinetic studies were carried out on male Sprague–Dawley rats by tail injection, including the distribution, metabolism and elimination of DTX-SBE-β-CD in vivo. The results revealed that the Cmax of DTX- SBE-β-CD was 1.8 times that of docetaxel API and the AUC (0-t) and AUC (0-∞) of DTX- SBE-β-CD were almost double that of Docetaxel API, which indicated that the formulation of docetaxel as a clathrate made with SBE-β-CD significantly promoted and sustained the absorption of docetaxel. In vivo distribution of the results shows that the distributions of DTX and DTX-SBE-β-CD in organs or tissues were almost the same and the changing trend was similar. Besides, hemolysis and vascular stimulation testing also demonstrated the safety of the DTX-SBE-β-CD.

In conclusion, our results demonstrated that the DTX- SBE-β-CD inclusion complex was formed by the saturated aqueous solution method and could improve the oral bioavailability and ability to inhibit tumor cells compared to DTX alone.

## Figures and Tables

**Figure 1 polymers-12-02336-f001:**
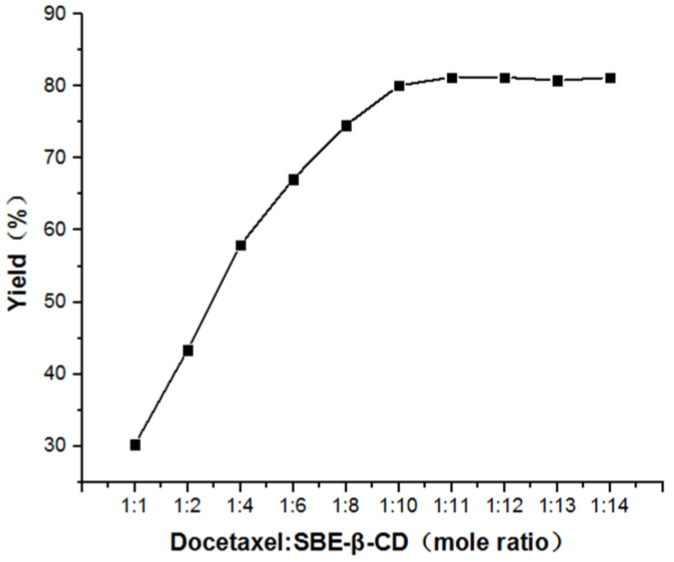
The molar ratio of docetaxel and sulfobutyl-ether-β-cyclodextrin (SBE-β-CD).

**Figure 2 polymers-12-02336-f002:**
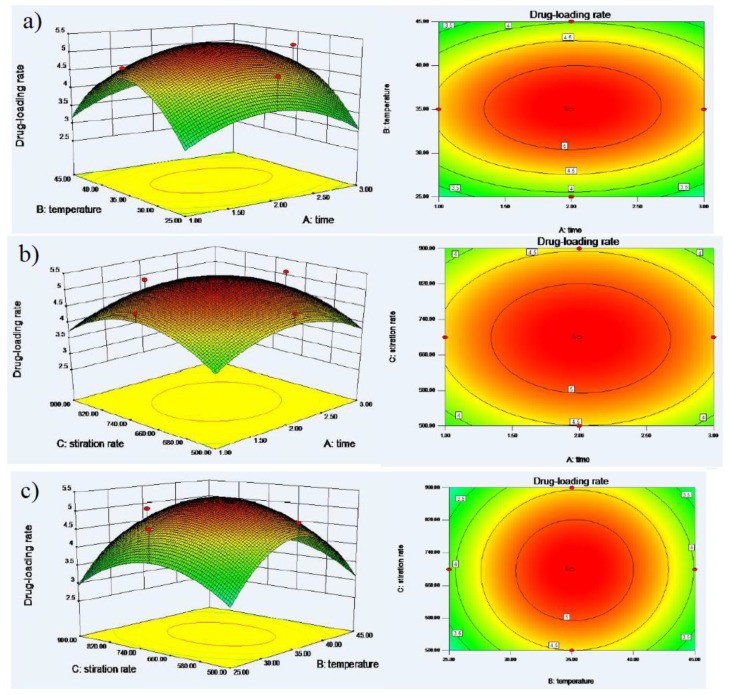
(**a**) Response surface plot (3D) and contour plot showing the effect of time (A) and temperature (B) added on the drug-loading rate. (**b**) Response surface plot (3D) and contour plot showing the effect of time (A) and stiration rate (C) added on the drug-loading rate. (**c**) Response surface plot (3D) and contour plot showing the effect of temperature (B) and stiration rate (C) added on the drug-loading rate.

**Figure 3 polymers-12-02336-f003:**
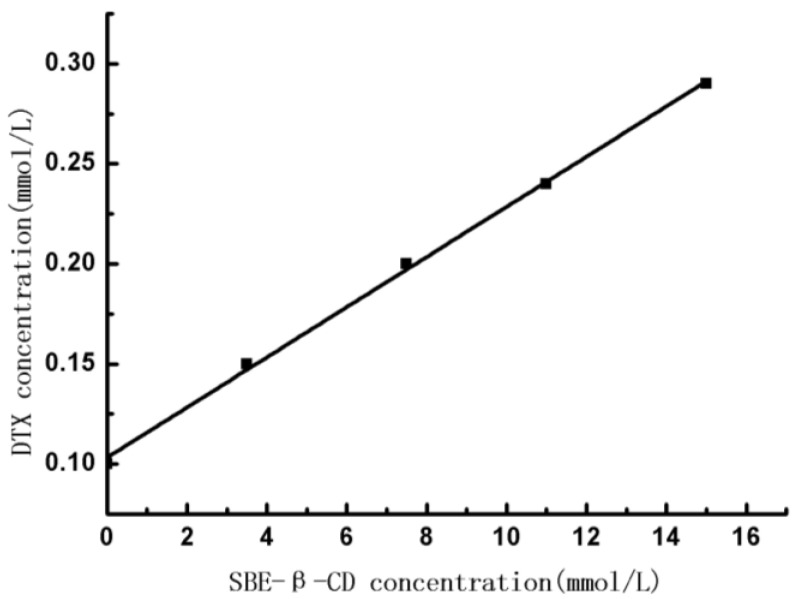
Docetaxel (DTX)/SBE-β-CD phase solubility diagram at 25 °C.

**Figure 4 polymers-12-02336-f004:**
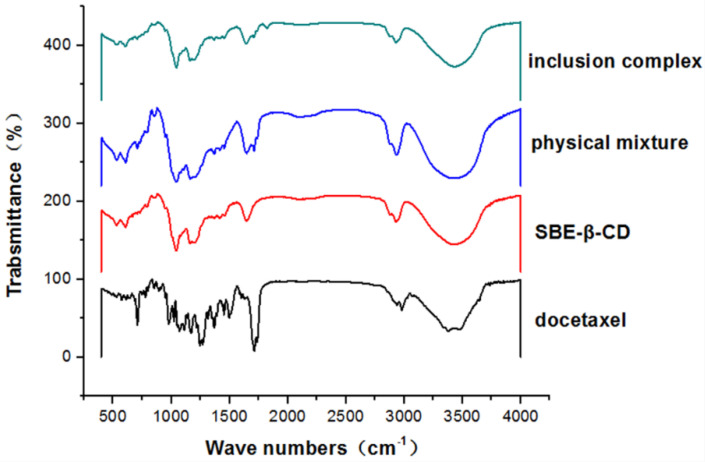
Infrared spectrogram of SBE-β-CD, docetaxel, physical mixture of docetaxel and SBE-β-CD, inclusion complex.

**Figure 5 polymers-12-02336-f005:**
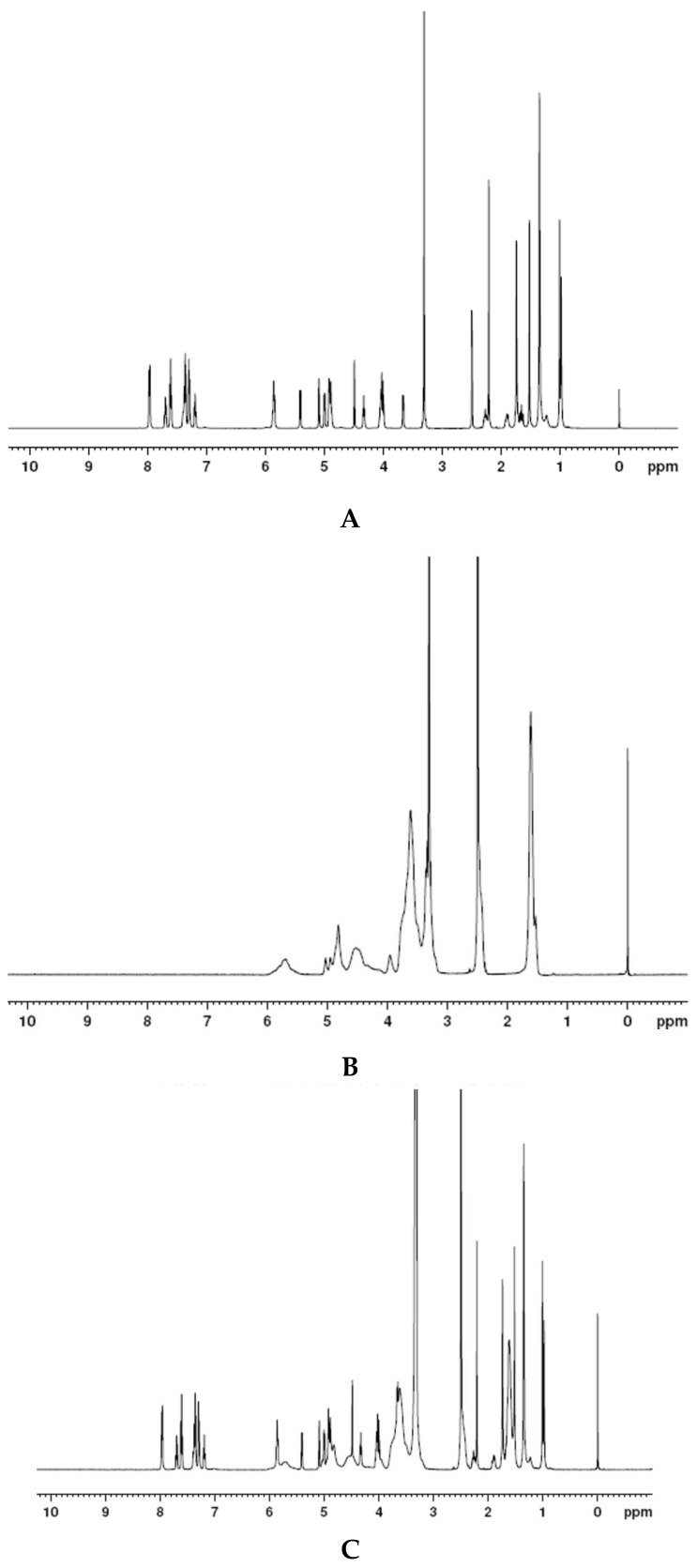
^1^H NMR photographs of (**A**) docetaxel; (**B**) SBE-β-CD; (**C**) physical mixture of docetaxel and SBE-β-CD; (**D**) inclusion complex of docetaxel and SBE-β-CD.

**Figure 6 polymers-12-02336-f006:**
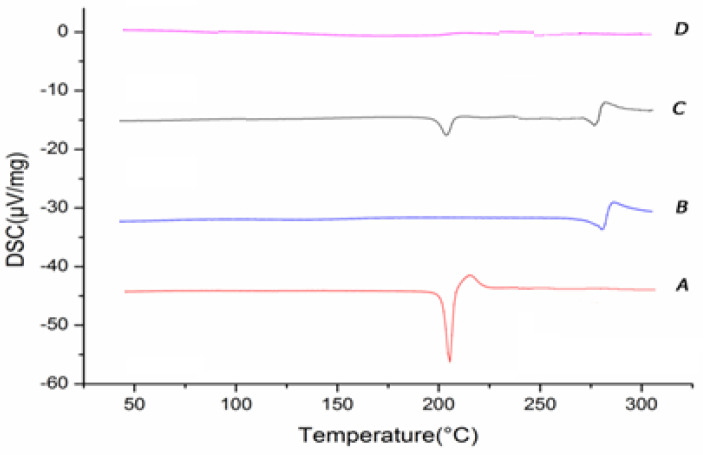
Differential scanning calorimetric diagrams (**A**) docetaxel; (**B**) SBE-β-CD; (**C**) physical mixture of docetaxel and SBE-β-CD; (**D**) docetaxel-SBE-β-CD inclusion complex.

**Figure 7 polymers-12-02336-f007:**
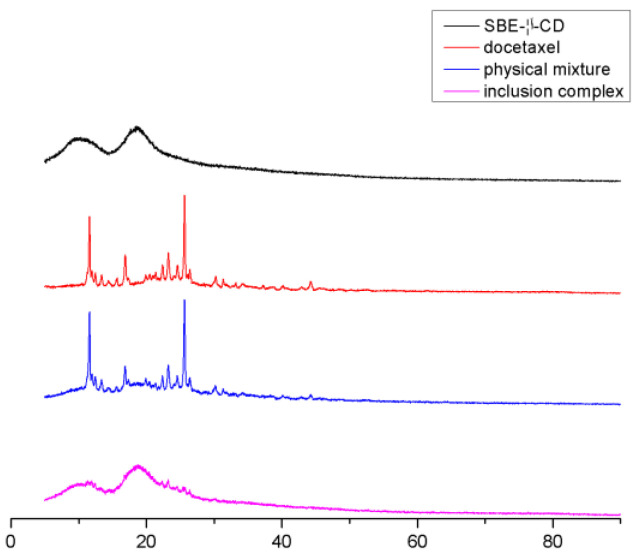
X-ray diffraction spectra of SBE-β-CD, docetaxel, their physical mixture and DTX-SBE-β-CD inclusion complex.

**Figure 8 polymers-12-02336-f008:**
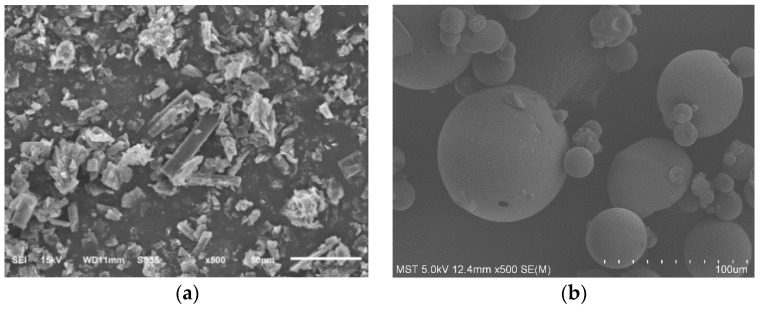
SEM photographs of (**a**) docetaxel; (**b**) SBE-β-CD; (**c**) physical mixture of docetaxel and SBE-β-CD; (**d**) inclusion complex of docetaxel and SBE-β-CD.

**Figure 9 polymers-12-02336-f009:**
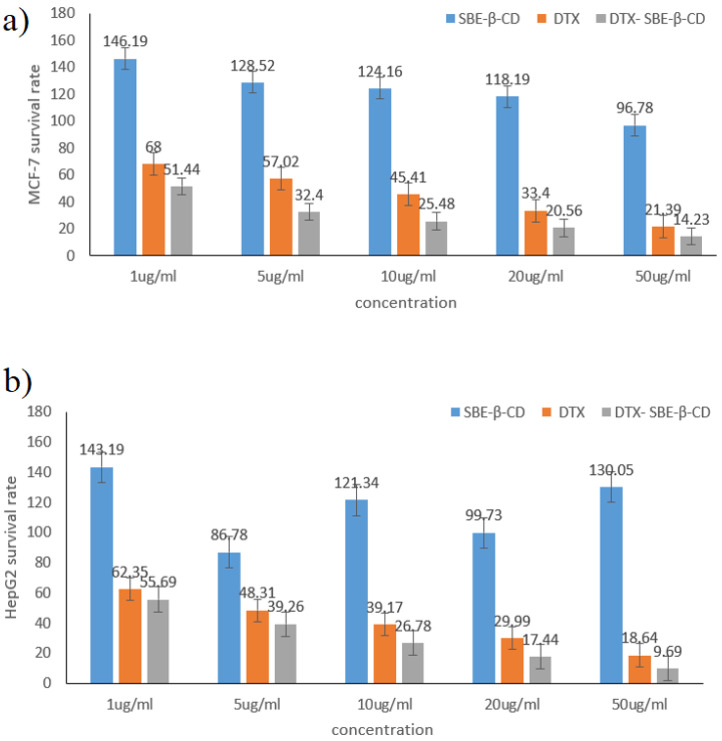
The survival rate of SBE-β-CD, DTX and DTX- SBE-β-CD on two human cancer cells: (**a**) MCF-7, (**b**) HepG2.

**Figure 10 polymers-12-02336-f010:**
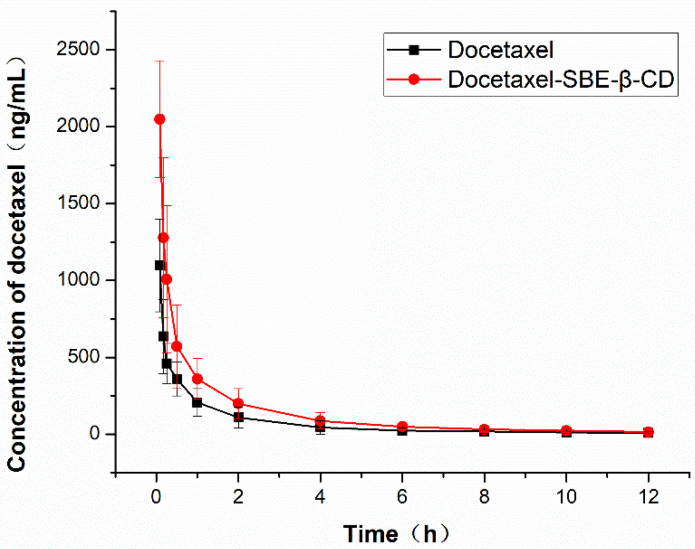
Pharmacokinetic parameters of docetaxel and DTX-SBE-β-CD in rats (*n* = 12, 10 mg/kg).

**Figure 11 polymers-12-02336-f011:**
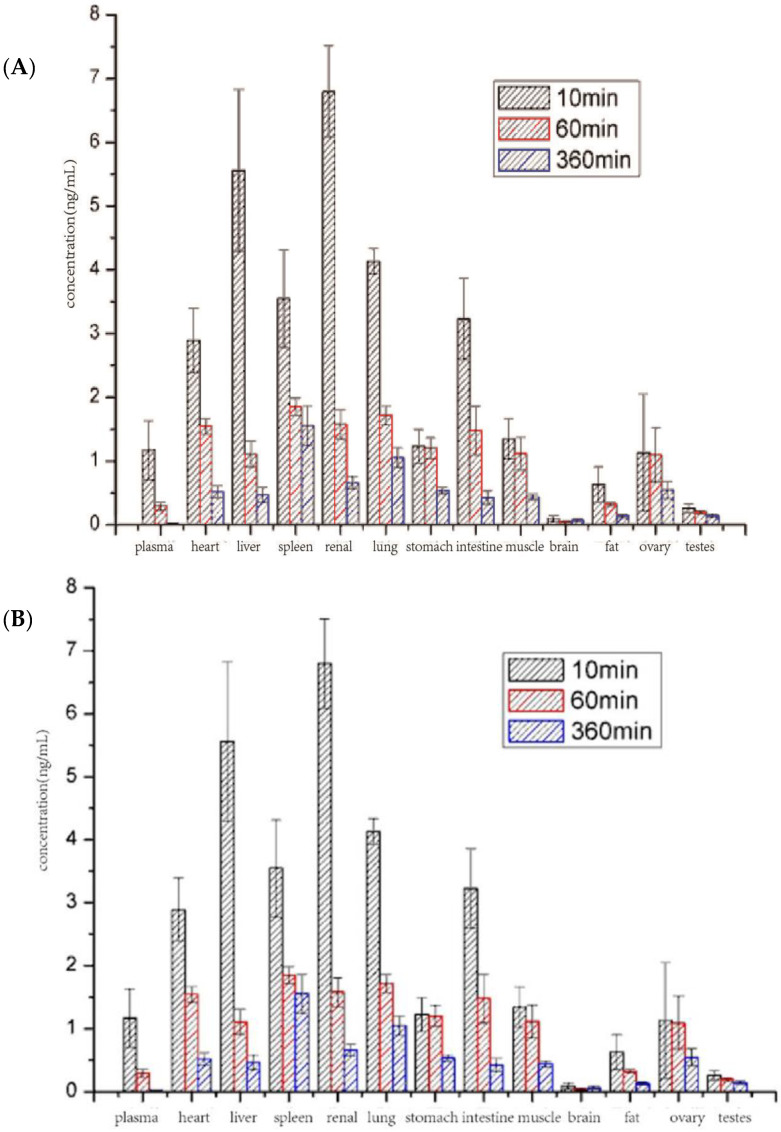
Biodistribution of DTX (**A**) and DTX-SBE-β-CD (**B**). After administration, DTX accumulation in organs or tissues were measured after 10, 60, and 360 min.

**Figure 12 polymers-12-02336-f012:**
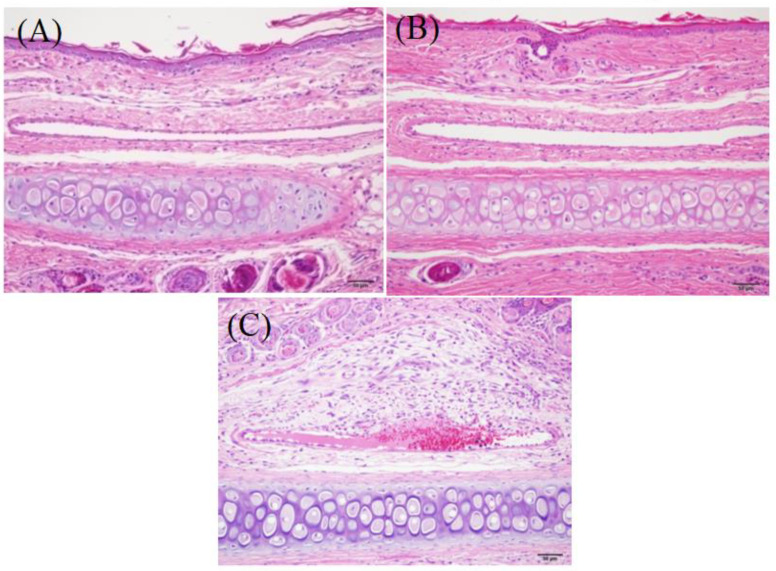
The micrograph of rabbit vein after mainlining. (**A**) Physiological saline; (**B**) the solution of docetaxel-SBE-β-CD; (**C**) docetaxel injection.

**Table 1 polymers-12-02336-t001:** The ratios of reagents in the hemolytic test of docetaxel freeze-dried powder injection. Red blood cell suspension (A), saline (B), testing sample (C).

Number	1	2	3	4	5	6	7
A	2.5	2.5	2.5	2.5	2.5	2.5	2.5
B	2.4	2.3	2.2	2.1	2.0	2.5	2.5
C	0.1	0.2	0.3	0.4	0.5	—	—

**Table 2 polymers-12-02336-t002:** Experimental factors and response variable in CCD for docetaxel-sulfobutyl ether-β-cyclodextrin inclusion complex (DTX-SBE-β-CD).

Experimental Factors	Levels					
	−1	0	1			
A (h)	1	2	3			
B (°C)	25	35	45			
C (rpm)	500	700	900			
Run	A			B	C	Drug-loading rate (%)
F1	2			25	700	4.72
F2	2			35	700	5.02
F3	1			25	900	2.1
F4	2			35	500	4.85
F5	3			45	500	2.48
F6	2			35	900	4.85
F7	2			35	700	5.02
F8	2			35	700	5.02
F9	2			45	700	3.85
F10	3			45	900	2.45
F11	3			25	500	2.07
F12	3			35	700	5.1
F13	2			35	700	5.12
F14	2			35	700	5.0
F15	3			25	900	1.9
F16	1			45	500	2.36
F17	1			35	700	4.85
F18	1			25	500	2.14
F19	1			45	900	2.4
F20	2			35	700	5.26

**Table 3 polymers-12-02336-t003:** Pharmacokinetic parameters of docetaxel in rats (*n* = 12) after injection of docetaxel API and DTX-SBE-β-CD.

	Docetaxel API	DTX-SBE-β-CD
Cmax (ng/mL)	1096.9 ± 98.0	2049.2 ± 147.4
Tmax (h)	0.083	0.083
AUC(0-t) (ng/mL ∗ h)	881.4 ± 30.4	1629.0 ± 34.3
AUC(0-∞) (ng/mL ∗ h)	925.8 ± 35.2	1700.6 ± 41.9
Relative bioavailability (%)	100	183.8

**Table 4 polymers-12-02336-t004:** The results of hemolysis test.

Time	1	2	3	4	5	6	7
0.25	−	−	−	−	−	−	+
0.5	−	−	−	−	−	−	+
0.75	−	−	−	−	−	−	+
1	−	−	−	−	−	−	+
2	−	−	−	−	−	−	+
3	−	−	−	−	−	−	+
4	−	−	−	−	−	−	+

Note: “+” means hemolysis; “−” means anhemolysis.

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
