# Peer review of "Inclusion Complex of Docetaxel with Sulfobutyl Ether β-Cyclodextrin: Preparation, In Vitro Cytotoxicity and In Vivo Safety"

_polymers, 2020, doi:10.3390/polym12102336_

Round 1
Reviewer 1 Report
In this work, the inclusion complex formation between sulfobutyl ether b-cyclodextrin and docetaxel was investigated with the aim to improve the solubility of the anti-tumour drug. Although the chosen methods can be used to analyze the investigated phenomena of the present work, the manuscript is completely incomprehensible. Several references are missing from the Introduction which can support the statements of this section (e.g. line 37 “… the complexation with cyclodextrins…”). Based on the Experimental part neither can be understood nor reproduce the presented work. The parts of Materials and Methods are mixed (e.g. section 2.3 must be transferred to methods), the setups of the measurements are incomplete (e.g. section 2.4.3). In Section 3., the discussion of the results is also incomplete (e.g. section 3.1). In general, poor language prevents the reader from understanding what the work has to say. Therefore, the manuscript in the present form supports the statements of neither the abstract nor the conclusion. However, based on the presented tables and figures I suggest resubmitting the manuscript after it has been fully rewritten and it has completed the statistical analyses of the presented in vitro and animal studies. Furthermore, I suggest deleting the computer simulation part. Or the calculation must be extended and compare e.g. the experimental and the theoretical NMR data etc.
Reviewer 2 Report
The study entitled “Inclusion complex of docetaxel with sulfobutyl ether β-cyclodextrin: preparation, in vitro cytotoxicity and in vivo safety” describes the preparation and characterization of docetaxel-SBE-βCD inclusion complexes as well as pharmacokinetics and inhibition effects of samples on cancer cells.
The study is not well designed. Some important aspects in relation to the preparation and characterization of samples must be taken into account in order to improve the quality of study as mentioned below.
2.4 Methods
The authors describe the preparation of DTX-SBE-βCD inclusion complexes by two different methods such as film hydration and freeze-drying. I have serious doubts that the first one can lead to the inclusion of drug into the cavity of CD. It is not a usual method employed in the preparation of inclusion complex. Other striking aspect is the high amount of cyclodextrin employed (1:12 ratio molar). This point must be clarified.
With respect to the freeze-drying method, DTX-SBE-βCD clathrate is employed. This point must be clarified including the amount of drug and polymer employed.
2.4.2 Central composite design
One factor affecting the preparation of inclusion complexes mentioned is the reaction time. The reaction that takes place must be described. Why has the molar ratio drug-polymer not been taken into account in this design?
2.4.4 Investigation of solubilization effect of DTX-SBE-βCD
In this regard, the well-known solubility diagram design (T. Higuchi, K. A. Connors, “Phase Solubility Techniques,” Advanced Analytical Chemistry of Instrumentation, Vol. 4, 1965, pp. 117-212) should be done in order to obtain information about the drug solubility in presence of increasing CD concentrations. In addition, the type of diagram obtained for DTX-SBE-βCD system (AL; BS as examples) would provide information about the stoichiometry of complex. In figure 1, yield is related to dissolution data and this is an important mistake. A solubility diagram must be designed as mentioned.
2.5 Characterization
In this study, FT-IR, H NMR, DSC and SEM techniques are employed for characterization of complexes. This point is so confusing because it is not clear which samples are characterized (film hydration, freeze dried samples or both). In addition, authors mention that DTX-SBE-βCD = 1:3, molar ratio (line 121) were characterized whereas FT-IR studies were carried out employing DTX-SBE-βCD (1:12) (line 125) and for the other techniques the samples analyzed are not mentioned. Therefore, it is not enough clear the samples analyzed. This important point must be clarified.
I highly recommend the use of X-ray diffraction technique (DRX) to check the inclusion of docetaxel in the SBE-βCD since it can be considered as the technique of choice in the study of inclusion complexes with CD, as is widely demonstrated in the bibliography. Crystalline drugs as docetaxel could adopt an amorphous state upon inclusion into the CD cavity. The information provided by other techniques (FT-IR, NMR or DSC, among others) would corroborate that obtained by XRD.
3.5.3 DSC study The sentence: the heat absorption peak of 215 meters (line 278) must be replaced by “the endothermic peak about 215° C. ”In the figure 5, the y-axis data (DSC (mV/mg) should be replaced by ΔH and endo and exo data applied.
Finally, the conclusions are written very briefly.
Reviewer 3 Report
In this work the inclusion complex of docetaxel with sulfobutyl ether β-cyclodetrin (DTX-SBE-β-CD) was prepared and optimized its properties to enhance the cytotoxicity of cancer cells. The work is interesting with considerable addition in the field and probably will earn wide interest which properties justify publication in Polymers. The references are upto date and comprehensive. The work is very nice and carefully discussed. I suggest publication after considering the following minor remarks:
Why the given derivative of BCD is chosen for these investigations ? Stereo-chemical reasons or simple size of the cavity ?
How the electronic densities of the BCD cavity affected by the sulfobutyl ether substituents ?
The different characterization methods applied at different temperatures. It is acceptable considering the narrow temperature range, but at least authors should estimate the binding constants between the species interacted. Drug loading rate is less informative and having these data thermodynamics of formation can also been followed.
How the mean substitution degree of 7 is considered during the molecular simulations ? The description of the molecular simulation results is very short.
Please insert the setup of the molecular simulation software (Discovery Studio 4.5) into the manuscript. (which method, which algorithm is used)
Please give the accessibility of Design-Expert.V8.0.6.1 on appropriate way as the journal request it (date, town).
Please enlarge the size of numbers on the scale of the NMR spectra.
I think, Figure 7 not diagram as it is cited in the caption. (line 316)
Reviewer 4 Report
This manuscript describes the preparation of inclusion complex of anticancer drug docetaxel with sulfobutyl ether beta-cyclodextrin and the examinatiom of cytotoxicity of prepared formulation. The experiments were well planned and the work is well written. I can see many improvements made after first revision. Some new analytical techniques were added as well as the conslusion of the work was developed. In my opinion the manuscipt can be published in Polymers.
Round 2
Reviewer 1 Report
The authors made effort to improve the quality of the manuscript and answered several questions, however, the statistical analyses of in vitro (Section 3.6) and pharmacokinetic (Section 3.7) and biodistribution (Section 3.8) studies are still missing. (Statistical significance, ANOVA, t test, Tukey’s post-hoc test …)
Reviewer 2 Report
After a second revision of this paper I consider that the authors have not taken into account some of my considerations.
In view of the new results presented (X ray diffraction pattern – figure 6) I considered that the complex is not formed. Firstly, the physical mixture does not show the profile of the cyclodextrin and secondly, the complex presents the main peaks of the drug.
When the authors are asked about the molar ratio employed in different techniques (DSC and FTIR) to evidence the obtaining of an inclusion complex, I am surprised that in both techniques the same sample has not been analyzed (author´s response: the ratio of the drug to the polymer used in the characterizations was 1:12. In the FT-IR, the ratio of the drug to KBr is 1:3)
For all these reason, I think that inclusion complex is not formed. Such a high content of cyclodextrin is not used for the inclusion of drug in the CD cavity and it only masks the results in the characterization of samples.
In view of the new data provided by the authors and the answers given to my questions, I consider that this work should not be published as it is presented due to it starts from the premise of the formation of an inclusion complex that is not formed.
Round 3
Reviewer 2 Report
From the answers again given by the authors, I have not changed my opinion.
I have serious doubts about the formation of a docetaxel-sulphobutyl-ether-beta-cyclodextrin inclusion complexes.